# The Results of Orthopaedic Medical Examinations in Adolescent Amateur Weightlifters

**DOI:** 10.3390/ijerph192113947

**Published:** 2022-10-27

**Authors:** Takuji Yokoe, Takuya Tajima, Nami Yamaguchi, Makoto Nagasawa, Yudai Morita, Etsuo Chosa

**Affiliations:** Division of Orthopaedic Surgery, Department of Medicine of Sensory and Motor Organs, Faculty of Medicine, University of Miyazaki, 5200 Kihara, Kiyotake, Miyazaki 889-1692, Japan

**Keywords:** adolescent weightlifters, orthopaedic medical examination, early sports specialization, gender difference

## Abstract

Few studies have reported injuries and physical characteristics in adolescent weightlifters. The purpose of this study was to report the results of orthopaedic medical examinations in adolescent amateur weightlifters that were performed cross-sectionally from 2012 to 2019. The orthopaedic medical examination included physical examinations, generalized joint laxity, muscle and joint tightness, static alignment, muscle volume of the lower extremities, and medial longitudinal arch of the foot (the height from the tip of the navicular tubercle to the ground surface). A questionnaire survey regarding pain in the spine and lower extremities was also performed. A total of 99 adolescent weightlifters were included (male/female, 71/28; mean age, 16.2 ± 0.2 years). A total of 9.1% had received orthopaedic treatments, with spine injuries being the most prevalent. Of those who had not received orthopaedic treatments, 31.1% had pain in the spine or lower extremities (for >4 weeks). There were no significant gender differences in the incidence of pain or positive findings of physical examinations. Female weightlifters had a more reduced dorsiflexion of the ankle joint than male weightlifters (*p* = 0.02). Male weightlifters had a lower flexibility of the quadriceps than female weightlifters. The results of orthopaedic medical examinations in this study may help clinicians and young weightlifters to prevent injuries in competitive weightlifting.

## 1. Introduction

Resistance training such as weightlifting has gained popularity among athletes because several studies have demonstrated that the incorporation of resistance training improves performance in sports activities [1,2,3]. However, weightlifting is associated with a high risk of low back pain (LBP), shoulder and knee injuries [4,5]. It remains controversial whether children and adolescent athletes should perform weightlifting as a part of training. In general, adolescent athletes are susceptible to sports-related injuries due to growth spurts and increased engagement in sports activities [6]. McCambridge et al. reported that weightlifting at the competition level was not recommended during puberty [7].

Several studies have reported on injuries in adult elite weightlifters [8,9,10]. According to a recent systematic review, the injury incidence among Olympic level weightlifters was 2.4–3.3 injuries/1000 h of training [8]. However, there is a paucity of studies evaluating the prevalence of injuries and physical characteristics in young amateur weightlifters. To the best of our knowledge, only one study has evaluated imaging findings in adolescent weightlifters [11]. Bush et al. found that serious injuries occurred more frequently in elite weightlifters specializing at younger ages (<16 years old) [12]. It was also shown that single-sport specialized young athletes in individual sports had a higher proportion of overuse injuries than those in team sports (44.3% vs. 32.2%, odds ratio = 1.67) [13]. Therefore, more studies regarding the incidence of injuries and physical characteristics of adolescent amateur weightlifters will be needed to evaluate the influence of early sports specialization on young weightlifters, as well as to protect those athletes and develop their talents. Moreover, during puberty, there are notable differences in sports-related injuries between male and female athletes. Stracciolini et al. reported that young female athletes had a higher percentage of overuse injuries, while young male athletes had a higher percentage of traumatic injuries [14]. Many causes for male–female discrepancies in the type of sports-related injuries have been described, including biomechanical [15,16], flexibility-associated [17], and anatomical and hormonal differences [18,19]. However, no studies have evaluated the gender differences in adolescent amateur weightlifters. Yokoe et al. reported the importance of orthopaedic medical examination (orthopaedic screening) for young amateur athletes as a screening tool for injury prevention and opportunity to educate athletes and their staff [20]. The purpose of this study was: (1) to report the results of orthopaedic medical examinations in young amateur weightlifters and (2) to evaluate the gender differences in the findings.

## 2. Materials and Methods

This study was approved by the institutional review board of our hospital (Accession No 2015-101). The study was ethically conducted according to international standards [21]. The written informed consent was obtained from a patient and his or her parent and/or legal guardian.

### 2.1. Participants

The records of orthopaedic medical examinations for young amateur athletes from 2012 to 2019 were retrospectively reviewed, and the results of adolescent weightlifters were identified. Orthopaedic medical examinations for young athletes were performed annually at a single institution in the study region. The details of the examinations have been described elsewhere [20]. All young amateur athletes included for orthopaedic medical examinations were junior high school or high school students from the study area who had been selected to participate in the annual national championship tournament (The Japanese national sports festival). This sports festival was first launched in 1946, and has been held annually thereafter. Among these young athletes, weightlifters were included in this study. The weightlifters who had been active in competitive weightlifting for less than one year were excluded from the analyses.

### 2.2. Procedures

The participants were questioned as to whether they had received an orthopaedic treatment and whether they had any pain or symptoms in the spine or lower extremities (for over 4 weeks) at the time of medical examinations. During the study period, orthopaedic medical examinations were performed by a total of 10 senior orthopaedic surgeons and 20 certified physical therapists who specialized in sports medicine. The orthopaedic medical examination was divided into six categories: category 1, physical examinations of the spine, knee, and ankle joints; category 2, the assessment of the generalized joint laxity (GJL); category 3, the assessment of the muscle and joint tightness of the lower extremities; category 4, the assessment of the static alignment of the lower extremities; category 5, the assessment of the muscle volume of the lower extremities; and category 6, the assessment of the height of the medial longitudinal arch of the foot. Category 1 was performed by orthopaedic surgeons, and other categories were performed by physical therapists. Each measurement (category 3–6) was performed 3 times and the mean of the obtained results was used for the analyses.

#### 2.2.1. Category 1

The physical examinations comprised the Kemp test for spine injuries [22], 6 assessments for knee injuries (lateral stress test of the patella, Lachman test, posterior drawer test, McMurray test, varus and valgus stress test), and 2 assessments for ankle injuries (anterior drawer test and inversion stress test). When the young athlete complained of any pain or disorders that could not be assessed by the above physical examinations, additional physical examinations were performed to evaluate them. 

#### 2.2.2. Category 2

The Tokyo University score for the assessment of GJL, which was introduced by Nakajima et al. [23], was used in the present study. This assessment method evaluates 6 bilateral joints (shoulder, elbow, wrist, hip, knee, and ankle joint) and the spine (Figure 1). When one of the 12 joints meets the criteria, a score of 0.5 point is given; when the spine meets the criteria, a score of 1.0 point is given. Total scores range from 0 to 7 points, with a higher score indicating the presence of GJL. 

#### 2.2.3. Category 3

Muscle and joint tightness of the lower extremities were evaluated by the finger floor distance (FFD) [24], Thomas test [25], straight leg raising (SLR) test [26], and Ely’s test [27], and an assessment of the active dorsiflexion of the ankle joint in the supine position with a goniometer.

#### 2.2.4. Category 4

The quadriceps angle (Q angle) was examined in the supine position with a goniometer to measure the line connecting the anterior superior iliac spine and the midpoint of patella intersecting with the line connecting the tibial tubercle to the midpoint of the patella [28]. The leg–heel angle was evaluated from behind the individual in a standing position, and was classified as supination, neutral, or pronation. 

#### 2.2.5. Category 5

The muscle volume of the lower extremities was assessed with the subject standing in a relaxed bipedal stance, according to the circumference of the thigh at 10 cm above the proximal tip of the patella, and the maximum circumference of the calf. 

#### 2.2.6. Category 6

The medial longitudinal arch of the foot was assessed by the height from the tip of the navicular tubercle to the ground surface with the subject standing in a relaxed bipedal stance [29]. A ruler was used to measure the height of the navicular tuberosity from the ground. Low arch was defined as a height shorter than 15 mm, as Roth et al. reported that the height of the navicular bone from the floor was in proportion with that of the longitudinal arch of the foot, and the mean navicular height in patients with flexible flat foot was 15.7 ± 4.3 mm [30].

### 2.3. Statistical Analyses

To compare the results of orthopaedic medical examinations between male and female weightlifters, statistical analyses were performed using the SAS software (JMP Pro ver. 15.2.0; SAS Institute, Cary, NC, USA). Descriptive statistics were reported as percentage or mean ± standard deviation (SD). The Shapiro–Wilk method was used to test whether the data were normally distributed. The chi-squared test or Fisher’s exact test was carried out for categorical variables. Student’s *t*-test was performed for continuous variables when the data showed a normal distribution, otherwise, the Mann–Whitney U test was performed. The significance of threshold was set at *p* < 0.05.

## 3. Results

A total of 99 adolescent weightlifters were included in this study. The characteristics of these weightlifters are summarized in Table 1. There were 71 male weightlifters (71.7%). The mean ages of male and female weightlifters were 16.2 ± 0.1 and 16.2 ± 0.2 years, respectively (*p* = 0.72). The mean duration of practicing weightlifting was 1.9 years (range, 1.0–2.5 years). The mean frequency and duration of each practice session/week were 6 times and 3 h, respectively. 

### 3.1. Adolescent Weightlifters Who Had Received Orthopaedic Treatment

A total of 9 weightlifters (9.1%) had received orthopaedic treatment at the time of orthopaedic medical examination (Table 2). Spine-related injuries accounted for 44.4% of the reasons why weightlifters had received orthopedic treatment. A higher percentage of female weightlifters had received orthopaedic treatment than male weightlifters, although there was no significant difference (14.3% vs. 7.0%, *p* = 0.27).

### 3.2. Results of Orthopaedic Medical Examinations in Young Weightlifters Who Had Not Received Orthopaedic Treatment

To exclude the influence of injuries on the results of examinations, we separately evaluated the findings in weightlifters who had not received orthopaedic treatment at the time of orthopaedic medical examination (n = 90).

#### 3.2.1. Category 1 

A total of 28 weightlifters (31.1%, 19 males and 9 females) had pain in the spine or lower extremities (Table 3). There was no significant difference in the prevalence of pain relating to the spine or lower extremities between male and female weightlifters (3.0% vs. 4.2%, *p* = 0.45). The origin of pain was the low back in 20 (22.2%), followed by the knee in 8 (8.9%), and the ankle in 3 (3.3%).

A total of 28 weightlifters (31.1%, 19 males and 9 females) had at least 1 positive finding at the physical examination of the spine or lower extremities (Table 4). There was no significant difference in the prevalence of positive findings between male and female weightlifters (3.0% vs. 4.2%, *p* = 0.45).

#### 3.2.2. Categories 2–6

The results of categories 2–6 are shown in Table 5. There were statistically significant differences between male and female weightlifters in Ely’s test (R) (57.6% vs. 25.0%, *p* = 0.008), Ely’s test (L) (53.0% vs. 25.0%, *p* = 0.03), dorsiflexion of the ankle (R) (15.2 ± 1.1° vs. 9.8 ± 1.9°, *p* = 0.03), dorsiflexion of the ankle (L) (14.7 ± 1.1° vs. 9.4 ± 1.9°, *p* = 0.02), and Q angle (R) (10.6 ± 0.5° vs. 13.3 ± 0.8°, *p* = 0.02). There were also significant gender differences in the circumference of the thighs and calves. There were no significant differences in other findings.

## 4. Discussion

The main findings of the present study were as follows: (1) 9.1% of adolescent amateur weightlifters had received orthopaedic treatment, and spine-related injuries were the most frequently treated injuries (44.4%). There were no significant gender differences. (2) Of the adolescent weightlifters who did not receive orthopaedic treatments, 31.1% had pain in the spine or lower extremities that had lasted for >4 weeks, and 65% of weightlifters with LBP had positive findings on the Kemp test. There were no significant gender differences. (3) There were significant gender differences in the tightness of quadriceps and dorsiflexion of the ankle joint between male and female adolescent weightlifters.

Several studies have reported on the injuries among adult elite weightlifters, noting that the shoulder, spine, and knee were the most common injury locations [4,8,31]. Calhoon et al. reported that acute injuries accounted for 59.6% of injuries, while chronic injuries accounted for 30.4% [32]. According to a systematic review, the incidence of injuries in weightlifters was similar to that in other non-contact athletes [8]. To the best of our knowledge, the present study was the first and largest population study to evaluate injuries and physical characteristics in adolescent amateur weightlifters. Spine-related pain and injuries were the most prevalent in adolescent weightlifters, similar to findings in elite adult weightlifters [8,31,32]. Almost 20% of young weightlifters had LBP for >4 weeks, and 65% of those weightlifters with LBP had positive findings on the Kemp test. LBP is a common symptom in young athletes, and Micheli et al. reported that spondylolysis was the most common cause of LBP, accounting for 47% of cases [33]. It was reported that 91.7% of young weightlifters had abnormal findings of the spine on magnetic resonance imaging (MRI) at the end of a 3-year cohort study [11]. Sakai et al. reported that the bone healing rate was 93.8% in early-stage spondylolysis [34]. Therefore, screening of LBP, especially for the early detection of spondylolysis, may be needed to protect young weightlifters. It was reported that the prevalence of injuries and positive findings of physical examinations were the highest in weightlifting among the four sports activities [20]. Thus, orthopaedic medical examinations are recommended for screening injuries in adolescent weightlifters. 

The results of the present study demonstrated no significant gender differences in the incidence of injury, pain in the spine or lower extremities, or the findings of physical examinations. It was reported that young female athletes sustained more injuries to the spine and lower extremities than young male athletes due to a retrospective review of 2133 young athletes [14]. Some authors have found that female athletes had a greater risk of knee injuries than males [35,36]. The sexual differences in the incidence and type of sports-related injuries are influenced by a number of factors [15,16,17,18,19]. Especially during puberty, different growth patterns between the gender, including the bone mineral density, can influence on the sports related injuries [37]. Quatman et al. reported sexual differences in “weightlifting”-related injuries among non-weightlifters [5]. However, no studies have evaluated the gender differences in injuries among competitive weightlifters. In the present study, female weightlifters showed a significantly lower dorsiflexion of the ankle joints than male weightlifters. Restricted dorsiflexion of the ankle joint is associated with increased knee valgus, a decreased knee flexion angle, and increased ground reaction forces on landing [38,39,40]. Therefore, limited dorsiflexion of the ankle may be a risk factor for knee injuries, such as anterior cruciate ligament injury and patellofemoral pain [39,40]. Bell et al. reported that medial knee displacement during squatting was reduced by improving the dorsiflexion of the ankle joint [41]. Therefore, an intervention to improve the dorsiflexion of the ankle joint may be necessary to prevent injuries in female weightlifters. However, the current study was unable to draw conclusions concerning gender differences in weightlifters due to its retrospective design and limited population size. Further studies are thus needed to investigate gender differences in young weightlifters as well as adult weightlifters in order to individualize prevention and treatment strategies for weightlifting. 

Whether or not early specialization is required for weightlifters remains controversial. All the participants in this study were early specialized in weightlifting. Early sports specialization may be necessary for some sports, including gymnastics and figure skating, because the best performance is achieved before physical maturation [42,43]. However, early sports specialization is associated with an increased risk of overuse injuries, burnout, and depression [43,44,45,46]. Bush et al. reported that 68.8% of adult elite weightlifters did not think that early specialization was necessary to achieve elite status [12]. Furthermore, the American Academy of Pediatrics recommended that preadolescents and adolescents should avoid maximal lifting before physical and skeletal maturity [7]. More studies will be required to evaluate the influence of early specialization on physical and psychological problems in adolescent weightlifters.

There are several limitations to the present study. First, the orthopaedic medical examinations in this study did not evaluate the upper extremities. Shoulder injuries are common in weightlifters [4,5]; therefore, future studies will be required to evaluate the injuries and physical characteristics of upper extremities in adolescent weightlifters. Second, the number of adolescent female athletes included in this study was not large. However, few female weightlifters specialize in competitive weightlifting at an early age, so this may have been unavoidable. Third, the orthopaedic medical examinations were performed cross-sectionally (annually). Therefore, the influence of early specialization on the career of weightlifters was not evaluated. Despite these limitations, the present study provided useful information regarding young amateur weightlifters for clinicians and researchers, as well as weightlifters and their healthcare providers.

## 5. Conclusions

A total of 9.1% of adolescent amateur weightlifters had received orthopaedic treatments, with spine-related injuries being the most prevalent. Of the adolescent weightlifters who did not receive orthopaedic treatments, a third reported pain in the spine or lower extremities lasting for >4 weeks, accompanied by positive findings on physical examinations. There were no significant gender differences in the incidence of pain or findings of physical examinations, although female weightlifters had more reduced dorsiflexion of the ankle joints.

## Figures and Tables

**Figure 1 ijerph-19-13947-f001:**
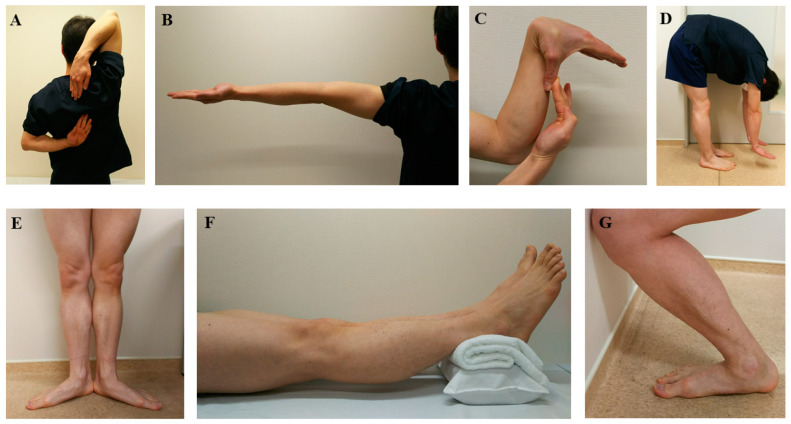
The Tokyo University score for generalized joint laxity. (**A**)**.** Crossing fingers behind the back. (**B**)**.** Hyperextension of both elbows beyond 15°. (**C**)**.** Passive opposition of both thumbs to volar aspects of ipsilateral forearms. (**D**)**.** Forward flexion of the trunk with the knees fully extended and palms resting on the floor. (**E**)**.** External rotation of both hips beyond 90°. (**F**)**.** Hyperextension of both knees beyond 10°. (**G**)**.** Weight-bearing ankle dorsiflexion range of motion beyond 45°. Tests other than D are bilateral. The patient receives a score for each individual joint, which is determined based on the evaluation of these items (score, 0 to 7).

**Table 1 ijerph-19-13947-t001:** Characteristics of adolescent weightlifters.

Variable	Male (n = 71)	Female (n = 28)	*p* Value
age, yrs	16.2 ± 0.1	16.2 ± 0.2	0.722
height, cm	167.1 ± 0.6	155.1 ± 1.0	<0.001
weight, kg	70.6 ± 1.4	57.4 ± 2.4	<0.001
BMI, %	25.2 ± 0.5	23.8 ± 0.9	0.157
The duration of competitive			
weightlifting	2.2 ± 1.3	1.8 ± 1.2	0.164

Data presented as mean ± standard deviation unless otherwise indicated. BMI, body mass index.

**Table 2 ijerph-19-13947-t002:** The reasons why adolescent weightlifters received orthopaedic treatments.

Causes	All (n = 99)	Male (n = 71)	Female (n = 28)	*p* Value
Spondylolysis	3 (3.0)	2 (2.8)	1 (3.6)	0.844
LBP	1 (1.0)	0 (0)	1 (3.6)	NA
Shoulder pain	1 (1.0)	1 (1.4)	0 (0)	NA
OCD of the elbow	1 (1.0)	1 (1.4)	0 (0)	NA
Jumper knee	2 (2.0)	1 (1.4)	1 (3.6)	0.488
Groin pain	1 (1.0)	0 (0)	1 (3.6)	NA
Total	9 (9.1)	5 (7.0)	4 (14.3)	0.266

Data presented as number (%). LBP, low back pain; OCD, osteochondritis dissecans; NA, not assessed.

**Table 3 ijerph-19-13947-t003:** Details of pain detected in the spine and lower extremities.

The Location of Pain	All (n = 90)	Male (n = 66)	Female (n = 24)	*p* Value
Low back	20 (22.2)	14 (21.2)	6 (25.0)	0.766
Knee	8 (8.9)	6 (9.1)	2 (8.3)	0.911
Achilles tendon	1 (1.1)	0 (0)	1 (1.1)	NA
Ankle	3 (3.3)	2 (3.0)	1 (4.2)	0.791

Data presented as number (%). NA, not assessed.

**Table 4 ijerph-19-13947-t004:** Results of physical examinations of the spine and lower extremities (category 1).

Variables	All (n = 90)	Male (n = 66)	Female (n = 24)	*p* Value
Kemp test	13 (14.4)	10 (15.2)	3 (12.5)	0.752
Mc Murray test	3 (3.3)	2 (3.0)	1 (4.2)	0.791
Valgus stress test of the knee	3 (3.3)	2 (3.0)	1 (4.2)	0.791
Varus stress test of the knee	2 (2.2)	2 (3.0)	0 (0)	NA
Tenderness at patella tendon	1 (1.1)	1 (1.5)	0 (0)	NA
Tenderness at Achilles tendon	1 (1.1)	0 (0)	1 (4.2)	NA
Inversion stress test of the ankle	2 (2.2)	0 (0)	2 (8.3)	NA
Anterior drawer test of the ankle	8 (8.9)	6 (9.1)	2 (8.3)	0.911

Data presented as number (%). The value in each variable shows the number of patients with positive findings. NA, not assessed.

**Table 5 ijerph-19-13947-t005:** Results of the orthopaedic physical examinations (categories 2–6).

Categories	All (n = 90)	Male (n = 66)	Female (n = 24)	*p* Value
Category 2				
GJL	1.5 ± 1.1	1.3 ± 0.1	1.8 ± 0.2	0.111
Category 3				
FFD, n (%)	17 (18.9)	15 (22.7)	2 (8.3)	0.221
Thomas test (R), n (%)	13 (14.4)	12 (18.2)	1 (4.2)	0.172
Thomas test (L), n (%)	12 (13.3)	11 (16.7)	1 (4.2)	0.170
SLR (R), °	71.4 ± 13.1	71.6 ± 1.6	70.8 ± 2.7	0.905
SLR (L), °	73.1 ± 13.2	73.7 ± 1.6	71.3 ± 2.7	0.601
Ely’s test (R), n (%)	44 (48.9)	38 (57.6)	6 (25.0)	**0.008**
Ely’s test (L), n (%)	41 (45.6)	35 (53.0)	6 (25.0)	**0.030**
Dorsiflexion of the ankle joint (R), °	13.7 ± 9.4	15.2 ± 1.1	9.8 ± 1.9	**0.028**
Dorsiflexion of the ankle joint (L), °	13.3 ± 9.3	14.7 ± 1.1	9.4 ± 1.9	**0.023**
Category 4				
Q angle (R), °	11.3 ± 4.1	10.6 ± 0.5	13.3 ± 0.8	**0.018**
Q angle (L), °	11.4 ± 3.7	10.9 ± 0.4	12.7 ± 0.7	0.051
Leg heel angle (R), n (%)				0.615
Pronation	13 (14.4)	10 (15.2)	3 (12.5)	
Supination	7 (7.8)	4 (6.1)	3 (12.5)	
Neutral	70 (77.8)	52 (78.8)	18 (75.0)	
Leg heel angle (L), n (%)				0.417
Pronation	14 (15.6)	12 (18.2)	2 (8.3)	
Supination	8 (8.9)	5 (7.6)	3 (12.5)	
Neutral	68 (75.6)	49 (74.2)	19 (79.2)	
Category 5				
Circumferencial length of the thigh (R), cm	50.3 ± 4.1	51.1 ± 0.5	48.3 ± 0.8	**0.005**
Circumferencial length of the thigh (L), cm	50.4 ± 4.2	51.1 ± 0.5	48.8 ± 0.8	**0.021**
Circumferencial length of the calf (R), cm	37.2 ± 3.4	38.0 ± 3.3	35.1 ± 2.7	**<0.001**
Circumferencial length of the calf (L), cm	37.4 ± 3.8	38.1 ± 0.4	35.4 ± 0.7	**<0.001**
Category 6				
Low medial longitudinal arch of the foot (R), n (%)	40 (44.4)	28 (42.4)	12 (50.0)	0.633
Low medial longitudinal arch of the foot (L), n (%)	40 (44.4)	27 (40.9)	13 (54.2)	0.339

Data presented as mean ± standard deviation unless otherwise indicated. The value in FFD, Thomas test and Ely’s test shows the number of patients with positive findings. BMI, body mass index; GJL, generalized joint laxity; FFD, finger floor distance; SLR, straight leg raising; R, right; L, left. Bolded *p* value indicates statistically significant difference between the groups (*p* < 0.05).

## Data Availability

The data presented in the present study are available on request from the corresponding author.

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
