# Peer review of "The Results of Orthopaedic Medical Examinations in Adolescent Amateur Weightlifters"

_ijerph, 2022, doi:10.3390/ijerph192113947_

Round 1
Reviewer 1 Report
Overall impression
This is a descriptive study that has the significance of reporting on data about adolescent weightlifters. This makes the study interesting but on the other hand, its external validity is limited since the study's population is very specific. So unless, the authors could make an association between weightlifting and resistance training with regards to how some knowledge from weightlifting could be transferred to resistance training, which is systematically being used as part of training in adolescent athletes, it might not attract a general audience.
General comments
The statistical analysis used in the study is the basic one. However, it would be more interesting if a complimentary analysis as for example an odds ratio probability analysis was carried out, considering data like the range of practicing weightlifting (1 to 2.5 years) that might be of significance.
Specific comments
Lines 15-16: this needs to be specified, as it actually is an evaluation of the MLA of the foot.
Line 20: Here and elfesewhere in the manuscript, it is reported "...sexual differences". I do not agree this statement and I recommend correcting to sex differences.
Line 72: "...annual national tournament" and then line 73: what kind of tournament was, as it is inferred that there were not only weightlifters there. It is suggested that it is more clearly described.
Section 2.2. Procedures: if understood correctly, the study did not have any inclusion or exclusion criteria. May be this should be reported and authors give a very short justification of why they choose to not have any.
Lines 75-77: It is suggested to reverse the order of the sentences. if a participant experienced pain or symptoms, then they would seek orthopaedic treatment. Correct or not?
The procedures have been described in an earlier study of this group, but some important information appears to be missing. For example, lines 123-125, for the anthropometric measurements of circumference a more detailed description is required. For instance, how many trials per measuring site were taken and what did authors do in the case of a difference over 0.5 cm. Also, how many measurements per outcome measure on each side? How were the data treated afterwards for the statistical analysis? was it the averaged value of each outcome measure or what?
Lines 147-149: since the secondary purpose is to examine the possible effect of sex, it should be added as a characteristic in Table 1, the mean frequency and duration separately for male and female athletes.
Table 1: the measurement units for BMI need to be reported.
Reviewer 2 Report
Overall: A well written paper on the study of ortopaedic medical examinations in adolescent amateur weightlifters.
To improve your paper I have a few comments/recommendations:
- Material and Methods. Has consent been obtained from subjects involved in the study? Please describe what procedure was used?
- Results: In this section appeared only descriptive statistics and t-test. What about results analysed with Chi -Square or Mann Whitney U. In statistical analyses subsection you mention its.
In the table 1 at the variable BMI please insert measured in %.
- Specific comment: for a good understanding of the text, please replace sexual differencies with gender differencies. Lines: 20 55, 60, 180, 187, 190, 191, 214, 219, 224, 235, 265.
